# Thermorheological Behavior of *κ*-Carrageenan Hydrogels Modified with Xanthan Gum

Pietro Renato Avallone [1,2], Simona Russo Spena [1], Stefano Acierno [2], Maria Giovanna Esposito [1], Andrea Sarrica [3], Marco Delmonte [3], Rossana Pasquino [1] and Nino Grizzuti [1,*]

1 Department of Chemical, Materials, and Industrial Production Engineering, Federico II University, P.le Tecchio 80, 80125 Naples, Italy
2 Department of Engineering, University of Sannio, Piazza Roma 21, 82100 Benevento, Italy
3 Perfetti Van Melle, 20045 Lainate, Milan, Italy
* Correspondence: nino.grizzuti@unina.it

**Abstract:** Hydrocolloids are long-chain biopolymers that can form viscous solutions or gels when dissolved in water. They are employed as rheological modifiers in various manufacturing processes or finished products. Due to its unique gelation properties, animal gelatin is one of the most widely used hydrocolloids, finding applications in several fields such as food, pharmaceutical, and photographic. Nowadays, the challenge of finding valid alternatives to animal products has become a crucial issue, for both ethical and environmental reasons. The aim of this work, is to propose a green hydrocolloidal network, able to reproduce the gelation features of animal gelatin gels. *κ*-carrageenan gels may be an interesting alternative to gelatin, due to their attractive gelling features. We investigate the thermorheological behavior of *κ*-carrageenan aqueous solutions at various concentrations, focusing on gel features such as transition temperature and gel strength. To improve the viscoelastic response of such gels, we add a viscosity-enhancing hydrocolloid, i.e., xanthan gum. The results show that the gel strength increases exponentially with xanthan concentration, thus suggesting a synergistic interaction between the two networks. We also study the effect of sucrose on the thermal and mechanical properties of modified gels, finding a marked increase in transition temperatures and gel elasticity. In recent years, three-dimensional (3D) food printing has been extensively studied in the food industry, due to its many advantages, such as customized food design, personalized nutrition, simplified supply chain, and the expansion of available food materials. In view of this growing interest for additive manufacturing, we also study the printability of the complete formulation composed of *κ*-carrageenan, xanthan gum and sucrose.

**Keywords:** green hydrocolloids; *κ*-carrageenan; xanthan gum; sol-gel transition; viscoelastic properties





## 1. Introduction

Hydrocolloids are defined as long-chain biopolymers that form viscous solutions or gels when dissolved in water [1]. Such biopolymers can be extracted from animals, seaweeds, and trees, or they can be obtained as products from fermentation processes [2]. They contain several hydroxyl groups, which determine their hydrophilicity, making them highly soluble in water. The solutions of most common hydrocolloids in food applications undergo a gelation process, and this property is employed to confer consistency and texture to the final products [3–5]. The gel formation involves the associations of dispersed polymer strands in solution, in such a way as to yield a three-dimensional network [6]. The associated regions, also known as junction zones, can be formed by two or more polymer chains, and through such junctions the gelation process takes place [7]. The three main gelation mechanisms of hydrocolloids proposed in the literature are cold-set, heat-set, and ionotropic gelation [8]. Biopolymers, like gelatin, agar, and *κ*-carrageenan, undergo a cold-set gelation, i.e., they create gels by decreasing temperature [9–11].

In the last decade, the challenge of finding valid alternatives to animal products has become a crucial issue, for both ethical and environmental reasons. In recent years, various gelatin replacements for the food industry have been considered [12–14], showing the possibility of using mixed gelling systems from plant-based sources such as pectin, agar, konjac glucomannan, $\kappa$-carrageenan, etc. Complex mixtures of different hydrocolloids or hydrolyzed fractions are necessary, in order to obtain specific features of the final product [14]. $\kappa$-carrageenan derives from red seaweed and is made up of alternating 3-linked $\beta$-D galactose 4-sulfate and 4-linked 6-anhydro-$\alpha$-galactopyranose repeating units, bearing one negative charge on each sulfate group [15]. Aqueous solutions of $\kappa$-carrageenan undergo thermoreversible gelation, where the polymeric coils are linked to each other via hydrogen bonds, to form a 3D network composed of double helices [16]. The coil–helix transition of $\kappa$-carrageenan solutions is affected by several parameters, such as the presence of ions, pH, thermal history, and biopolymer concentration [11,17,18].

Song et al. [19], proposed a mixture of $\kappa$-carrageenan and carboxymethylcellulose (CMC), to replace gelatin in soft candies. Compared to pure $\kappa$-carrageenan gels, the authors found that the addition of CMC improved the textural properties, by enhancing the elasticity of the biopolymeric network and reducing brittleness in a specific concentration range.

Many studies have reported the synergistic interaction of the $\kappa$-carrageenan/konjac glucomannan mixture, which led to the formation of a promising gel with relevant tensile properties. These systems, however, present a high water syneresis [20,21]. Agoub et al. [13], suggested a hydrogel composed of xanthan (0.5 wt%) and konjac glucomannan (0.5 wt%), as a possible substitute for gelatin, observing that, by varying the pH between 3 and 7, the gels showed melting temperatures ranging between 25 °C and 40 °C, but exhibited low gel strength compared to gelatin gels [22–24]. On these bases, $\kappa$-carrageenan, or its mixtures with other hydrocolloids, represents an interesting alternative to gelatin.

The aim of this work, is to study the mechanical properties of $\kappa$-carrageenan gels, in such a way as to design a green hydrogel able to replace animal gelatin. Nevertheless, $\kappa$-carrageenan solutions are known to produce firm and brittle hydrogels [25,26]. Increasing the biopolymer concentration usually improves the gel elasticity, but it also increases the sol/gel transition temperature [11]. With the aim of improving the gel strength while reducing the effect on transition temperatures, we chose to study the influence of an additional hydrocolloid, namely, xanthan gum, known as a thickening agent, on $\kappa$-carrageenan solutions.

Xanthan gum is an anionic biopolymer, that consists of 1,4-linked $\beta$-D-glucose residues with a trisaccharide side chain, attached to alternate D-glucosyl residues. This ordered structure permits weak intermolecular associations, which results in solutions with very high zero-shear viscosities [27]. Xanthan gum solutions are shear thinning and exhibit a high degree of pseudoplasticity [28]. They undergo a conformational transition upon heating, which is associated with the change from the rigid rod-like ordered state at low temperatures, to a more flexible, disordered state at high temperatures, with a resulting decrease in viscosity [29]. The temperature at which the conformational transition takes place depends on the ionic strength of the solution and on some specific molecular features, such as the pyruvic acid and acetic acid moieties attached to the side chain of the xanthan gum backbone [30].

Since sugar is one of the most common ingredients in sweet confectionery, we also evaluate the effect of sucrose on the rheological properties of the designed vegetable hydrogel. It is already known that, for sugar concentrations up to 40%, the gel strength increases in polysaccharide networks; for higher sugar concentrations, on the other hand, a non-monotonic trend is found [31].

The gel system composed of $\kappa$-carrageenan–xanthan–sugar is tested not only from a rheological point of view, but also from that of extrusion-based 3D-printing. The latter is an emerging technology, with the potential to influence the food manufacturing sector [32]. Three-dimensional printing allows the fabrication of structures in a layer-by-layer pattern, created in pre-designed files. Several advantages of 3D food printing have been reported, such as the customization of food structures, the alteration of food

texture, and the use of various food sources. To improve printability in extrusion-based printing, a good understanding of the material's rheological properties is required. The food formulation should be easily extruded through a nozzle and, at the same time, possess enough mechanical strength to keep shape fidelity when printed. An ideal gel should exhibit well-defined rheological and thermal properties, in terms of viscosity, gel strength, gelation kinetics, and transition temperatures [33,34]. Thus, a deep understanding of the rheological properties of the formulation to be printed is crucial to achieving a successful printed shape. Some authors have investigated the effect of rheological properties on 3D printing behavior, highlighting the importance of the rheological features of inks, such as shear-thinning behavior and yield stress [35,36]. Liu et al. [33] studied the printability of systems based on $\kappa$-carrageenan–xanthan–starch, pointing out the beneficial effects of xanthan on such systems.

For thermo-responsive hydrogels, the gelation temperature is a key parameter from which the appropriate printing temperature can be determined. The choice of the optimal printing temperature plays a dual role. On the one hand, it affects the mechanical properties (e.g., viscosity or elasticity), which are strictly related to the shape fidelity of the printed object. On the other hand, it influences the gelation kinetics and, thus, the adhesion between layers.

## 2. Materials and Methods

### 2.1. Materials

$\kappa$-carrageenan ($\kappa$-c), in powder form, was purchased from Sigma-Aldrich, Italy. In agreement with several studies, the chemical composition of $\kappa$-c used in this work shows a large amount of calcium, potassium, and sodium [37,38]. In particular, it contains: $Mg^{2+} = 300$ μg/g, $Ca^{2+} = 11,500$ μg/g, $Na^{2+} = 1900$ μg/g, $K^{2+} = 83,100$ μg/g, and an amount of total ash equal to 81 μg/g.

Food-grade xanthan gum (XG) was purchased from Foxwood Industrial Park, United Kingdom. Food-grade sucrose was kindly supplied by the Perfetti Van Melle company. All reagents were used as received. Bi-distilled water was used to prepare solutions.

### 2.2. Sample Preparation

Aqueous $\kappa$-c solutions, at different concentrations, ranging between 0.1 and 3 wt%, were prepared by dispersing the powder in bi-distilled water at 80 °C, by using a magnetic stirrer at 360 rpm for 3 h. The obtained sample was then transferred to a glass bottle and stored in the refrigerator.

The preparation method for $\kappa$-c/XG solutions was similar to the protocol for the pure $\kappa$-c samples. The main difference was to first disperse the XG in bi-distilled water, using a magnetic stirrer, at 80 °C. Once a homogeneous XG solution was obtained, $\kappa$-c was slowly added, and the protocol for pure $\kappa$-c solutions, previously reported, was followed. For these solutions, the $\kappa$-c concentration was fixed at 1.5 wt%, and that of XG between 0.1–1.5 wt%.

$\kappa$-c/XG solutions with sucrose were prepared by keeping the $\kappa$-c and XG concentrations fixed at 1.5 wt% and 1.3 wt%, respectively. Sucrose, on the other hand, was varied between 5 and 40 wt%. These solutions were prepared by initially dissolving the hydrocolloids in hot water at 80 °C. Sucrose was then added, and the solutions were stirred at 80 °C for at least 3 h, to ensure complete dissolution and sample homogeneity. In addition, in this case, the solutions were transferred to a glass bottle and stored in the refrigerator.

Before each test, the solutions were kept in a closed vessel at 80 °C for 35 min, in order to remove any thermal history of the samples.

### 2.3. Rheological Measurements

Dynamic rheological measurements were performed on a Discovery Hybrid Rheometer 2 (TA Instruments, New Castle, DE, USA), in the linear viscoelastic regime, using 40 mm sandblasted parallel plates, with a 1 mm gap. The thermal expansion of the measuring tool was automatically accounted for during the cooling/heating ramps. To prevent sample

evaporation during measurements, a solvent trap and an oil sealing ($\eta_{oil} = 0.1$ Pa·s at 25 °C) were used.

Steady flow curves were carried out on a MCR 702 rheometer (Anton Paar, Graz, Austria), equipped with a Couette flow cell (CC 27). A thin top layer of silicon oil was added, to avoid sample evaporation. Both rheometers were stress-controlled, and equipped with a Peltier unit.

Dynamic strain sweep tests (DSSTs) were performed, at a fixed frequency of 10 rad/s and temperature of 20 °C, in an oscillatory strain range between 0.1 and 500%. This kind of test was used to evaluate both the linear viscoelastic regime and the yield stress of samples in the gel-like state. DSSTs were conducted by using 25 mm parallel plates, covered with an adhesive-backed waterproof sandpaper with a $25 \pm 1$ μm particle size, to prevent wall-slip phenomena [39].

Dynamic temperature ramp tests (DTRTs) were carried out at a frequency of 10 rad/s and a deformation of 5%, within the linear viscoelastic region. To determine the temperature dependence of the viscoelastic moduli, each DTRT was performed in a temperature range between 80 °C and 20 °C, at fixed cooling/heating rates (1 °C/min, 3 °C/min, and 5 °C/min). Hot solutions were loaded into the rheometer at 80 °C, cooled down to 20 °C, and, after a soaking time of 300 s, heated up again to 80 °C. The transition temperatures, defined as the minimum of the derivative of $log(|G^*|)$ with respect to temperature, were evaluated via DTRTs. The transition temperature during cooling is indicated as $T_{sol\text{-}gel}$, whereas the transition temperature during heating is indicated as $T_{sol\text{-}gel}$ [40,41].

Dynamic frequency sweep tests (DFSTs) were performed at 20 °C, with a linear strain of 5%, in a frequency domain between 100 and 0.1 rad/s.

The reproducibility of the rheological tests was verified through multiple measurements on fresh samples.

### 2.4. Inverse Vial Test

Small amounts of hot $\kappa$-c solutions at different biopolymer concentrations were directly transferred into 20 mL vials, and kept in a refrigerator at 5 °C for 24 h. After this aging time, the vials were turned upside-down at room temperature, to qualitatively assess the macroscopic gel formation [42].

### 2.5. 3D Printing Process

A 3D bioprinter (BIOX Cellink, Gothenburg, SE, USA), equipped with an extrusion-based and temperature-controlled printhead, was used to assess the printability of the hydrogels [43]. The bioprinter employs compressed air to extrude the solutions. The formulations were kept at 80 °C, without magnetic stirring, to avoid air bubble inclusions, and then poured into 3 mL cartridges to be inserted into the printer. Before the printing process, the formulations were held at different printing temperatures for 30 min. A nozzle of 0.41 mm diameter was used to perform the printing process.

The printing temperature was varied between 45 °C and 60 °C by the temperature-controlled nozzle. The bottom plate of the printer was controlled by a Peltier unit and kept fixed at 25 °C. The value of the printing speed was kept fixed and constant at 15 mm/s. Two-dimensional grid structures were extruded to evaluate the printability of the gels, in a printing pressure range between 4 and 10 kPa. Once the printing process was complete, pictures were captured using a 12 MP (f/1.5) camera (Samsung Galaxy S10, Seoul, Republic of Korea). The ImageJ software was used to assess the filament diameter of the printed structures.

## 3. Results and Discussion

### 3.1. Rheological Behavior of κ-Carrageenan Solutions

To characterize the viscosity evolution at different amounts of $\kappa$-c in solution, flow curves were performed at 60 °C in a Couette geometry. Figure 1a shows the results obtained

in steady shear flow for $\kappa$-c solutions at different concentrations, where we observe that the viscosity increases with increasing biopolymer amount, as expected.

By increasing the concentration of $\kappa$-c from 0.1 wt% to 3 wt%, there is an increase of three orders of magnitude in the zero shear viscosity. Newtonian behavior can be observed at low concentrations, whereas, at high biopolymer concentrations, a moderate shear thinning is shown [44]. The flow curves are also used to detect the concentration regimes, and to identify the overlap concentration, $c^*$, above which the solutions are in the semi-dilute regime [45].

Figure 1b reports the specific viscosity as a function of $\kappa$-c concentration. A power-law dependence of the viscosity on the concentration is observed, and two concentration regimes can be distinguished. Solutions in the concentration range 0.1–0.46 wt%, lying on the red regression line (in a log–log plot), with a slope of $\approx 0.92$, are in the dilute regime. Samples in the range of concentrations 0.7–3 wt% (black line, slope $\approx 2.42$), belong to a semi-dilute concentration regime [46]. The overlap concentration, determined by the intersection of the two regression lines shown in Figure 1b, is roughly 0.7 wt%. By using the value of $c^*$, it is possible to estimate the intrinsic viscosity, $[\eta]$, of the $\kappa$-c in water, via the relation $[\eta] \sim 1.45/c^*$ [47]. The estimated value of $[\eta]$ is 210 mL/g. The intrinsic viscosity of a polymer solution is related to the molecular weight $M$ by the Mark–Houwink equation:

$$[\eta] = KM^\alpha \tag{1}$$

where $K$ and $\alpha$ are tabled constants, available for polymer–solvent couples [48]. For $\kappa$-c, the values $K = 0.00598$ mL/g and $\alpha = 0.9$, reported in [49], are used, yielding a molecular weight of about $1.1 \times 10^5$ g/mol.

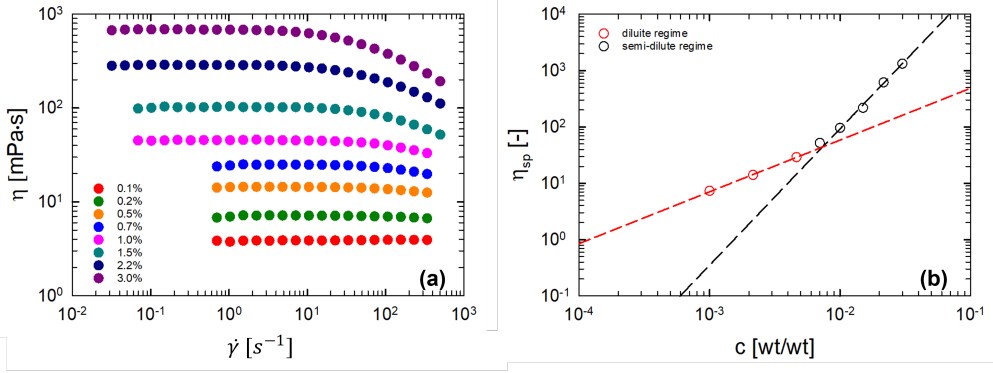

**Figure 1.** (**a**) Steady viscosity, $\eta$, as a function of shear rate, $\dot{\gamma}$, parametric in the $\kappa$-c concentration, at 60 °C. (**b**) Specific viscosities, $\eta_{sp}$, at 60 °C ($\eta_{water,60°C} = 5 \times 10^{-4}$ Pa · s) as a function of the $\kappa$-c concentration in solution. Power-law regressions are reported with dashed lines (red line, slope $\approx 0.92$; black line, slope $\approx 2.42$).

The thermorheological behavior of the $\kappa$-c solutions was analyzed through dynamic temperature ramp tests, performed in the linear viscoelastic region ($\omega = 10$ rad/s and $\gamma = 1\%$) at different heating/cooling rates. For solutions lying in the diluted regime, no thermal test was performed since no gelation occurs; these concentrations are below the critical concentration, $c_0$, needed to form a macroscopic gel [50,51].

The critical concentration was obtained by a vial inversion method [52], as shown in Figure 2. The critical concentration for $\kappa$-c solution is estimated to be 1 wt%, higher than the overlap concentration, in agreement with the findings of Clark and co-workers [50,51]. In the dilute regime, indeed, biopolymers form only finite aggregates, that cannot percolate through the solution, and give rise to the "infinite network" that characterizes the gel state.

Figure 3 shows the temperature evolution of the viscoelastic moduli during a DTRT performed at 1 °C/min on a $\kappa$-c solution at 1.5 wt%. As reported elsewhere [53,54], in the cooling phase, from 70 °C to 20 °C, $G''$ is initially higher than $G'$, and both moduli increase slightly with decreasing temperature, due to the decrease in molecular mobility of the

chains. This means that, at high temperatures, a liquid-like behavior is detected. With a further decrease in temperature, $G'$ and $G''$ increase abruptly, and after their crossover point $G'$ overcomes $G''$, marking the overcoming gelation phenomenon [11]. At low temperatures, close to $20\,°C$, a gel network is formed as a consequence of the formation and subsequent aggregation of helices, shaped by the association of adjacent polymer chain segments [55]. During the heating ramp, the opposite behavior is observed. Initially, the moduli slowly decrease as the temperature increases. Then, a sharp drop in moduli is observed, due to the melting of the 3D network, and a new crossover between the moduli occurs. After this characteristic point, $G''$ overcomes $G'$ and the viscoelastic moduli return to their initial values at high temperature. Thus, $\kappa$-c solutions exhibit typical thermoreversible behavior, although a clear hysteresis appears between the cooling and heating ramps.

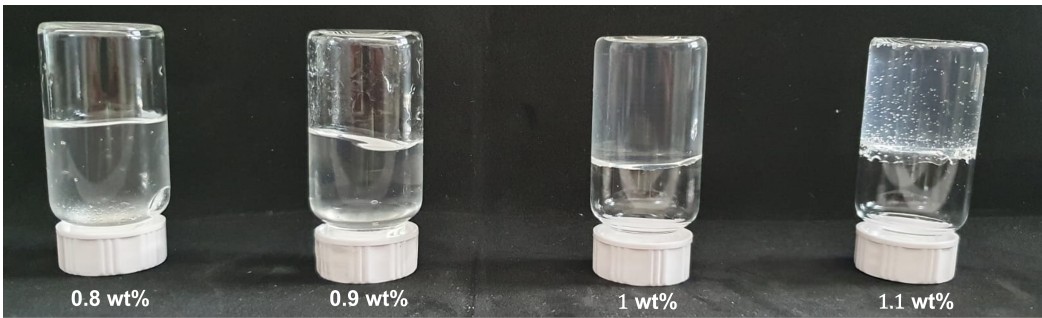

**Figure 2.** Inverse vial experiment. Visual observation of aqueous $\kappa$-c solutions at different biopolymer contents, as indicated in the legend. Picture is taken at room temperature, of samples kept in the refrigerator at $5\,°C$ for 24 h.

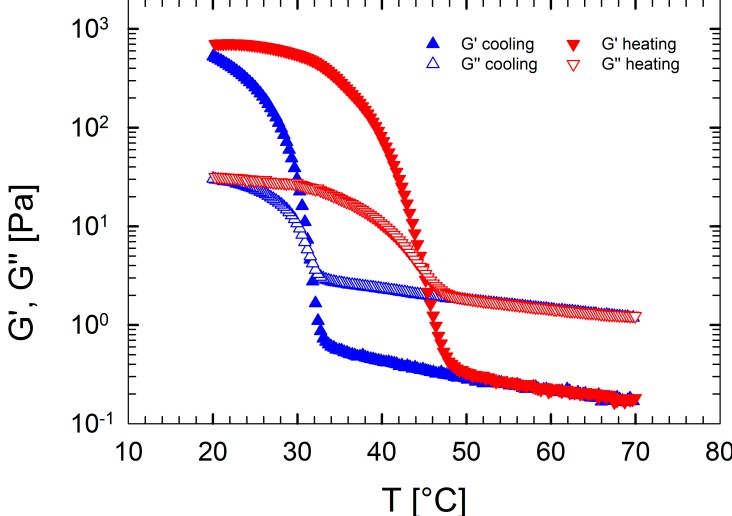

**Figure 3.** Storage modulus $G'$ and loss modulus $G''$, as a function of temperature, for an aqueous $\kappa$-c solution at 1.5 wt% and $1\,°C/\text{min}$. Blue up-pointing triangle symbols indicate the cooling phase, red down-pointing triangle symbols, the heating phase.

Figure 4 reports the transition temperatures as a function of the imposed cooling/heating rate, for solutions with $\kappa$-c concentrations ranging between 1.5 wt% and 3 wt%. The dashed lines are the best linear fits of the experimental transition temperatures, obtained via Equation (2):

$$T(r) = T_{r=0} \pm \tau \cdot r \tag{2}$$

where $T_{r=0}$ and $\tau$ are fitting parameters. The term $\pm$ is used to always obtain a positive value of the $\tau$ parameter. Thus, the minus sign is used for fitting the $T_{sol\text{-}gel}$ and the plus sign for the $T_{sol\text{-}gel}$ data. Fitting parameters obtained via Equation (2) are reported in Table 1.

As reported by Liu et al. [11], the fitting parameter, $T_{r=0}$, evaluated during the cooling ramp, can be assumed to be the equilibrium gelation temperature.

**Table 1.** Fitting parameters of Equation (2).

|  | $\kappa$-c Concentration | $T_{r=0}$ [°C] | $\tau$ [min$^{-1}$] |
|---|---|---|---|
| $T_{sol\text{-}gel}$ | 1.5 wt% | $32.0 \pm 0.7$ | $0.94 \pm 0.1$ |
| $T_{sol\text{-}gel}$ | 2 wt% | $36.1 \pm 0.2$ | $1.15 \pm 0.7$ |
| $T_{sol\text{-}gel}$ | 2.5 wt% | $39.1 \pm 0.7$ | $1.33 \pm 0.2$ |
| $T_{sol\text{-}gel}$ | 3 wt% | $40.5 \pm 0.04$ | $1.02 \pm 0.01$ |
| $T_{sol\text{-}gel}$ | 1.5 wt% | $43.1 \pm 0.5$ | $0.41 \pm 0.1$ |
| $T_{sol\text{-}gel}$ | 2 wt% | $48.8 \pm 0.3$ | $0.46 \pm 0.09$ |
| $T_{sol\text{-}gel}$ | 2.5 wt% | $52.5 \pm 0.3$ | $0.41 \pm 0.09$ |
| $T_{sol\text{-}gel}$ | 3 wt% | $56.4 \pm 0.3$ | $0.14 \pm 0.09$ |

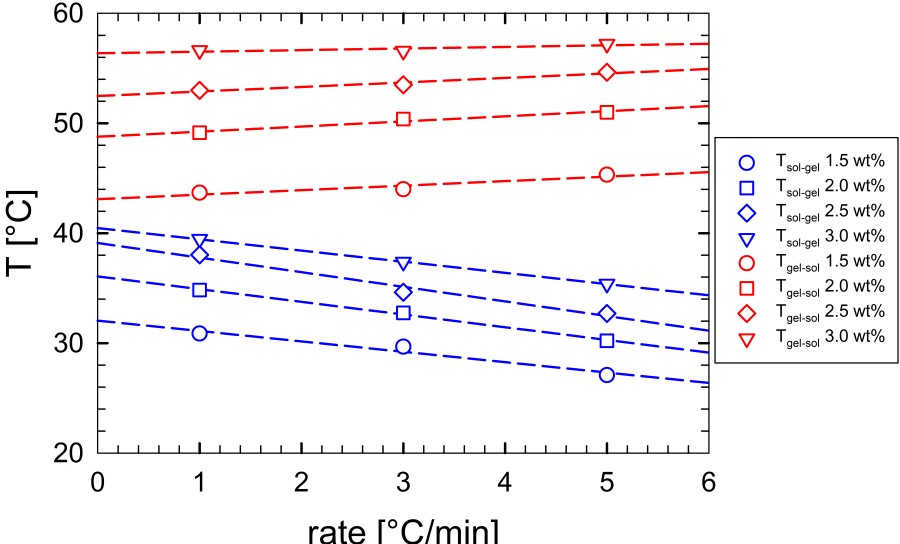

**Figure 4.** $T_{sol\text{-}gel}$ (blue symbols) and $T_{sol\text{-}gel}$ (red symbols) as a function of the imposed cooling/heating rate for different solutions (see legend for details). Dashed lines are linear regressions.

Figure 5 reports the sol-gel transition temperature, evaluated through the cooling ramp, and the gel-sol transition temperature, evaluated via the heating ramp, both at 1 °C/min, as a function of the $\kappa$-c concentration. In the gray region, that is, below the critical concentration of $c_0 = 1$ wt%, no gelation occurs.

In Figure 5, we observe a linear dependence of $T_{sol\text{-}gel}$ and $T_{sol\text{-}gel}$, as functions of the biopolymer concentration. The increase in $T_{sol\text{-}gel}$ with concentration is due to an enhanced formation and aggregation of double helices [55]. In addition, at high concentrations, larger aggregates are established in the 3D network; thus, more energy is required to destroy the helices and return to the coil conformation, which implies higher gel-sol temperatures [56]. The dashed lines are the best fits obtained via linear regression, whose parameters are $T_0$ and $a$ (reported in Table 2). The linear fits are empirical, and their validity is only proven for $\kappa$-c solutions in a semi-dilute regime.

**Table 2.** Values of the regression parameters of the linear fits in Figure 5.

|  | $T_0$ [°C] | $a$ [wt%$^{-1}$] |
|---|---|---|
| $T_{sol\text{-}gel}$ | $22.7 \pm 1.9$ | $5.8 \pm 0.8$ |
| $T_{sol\text{-}gel}$ | $31.1 \pm 2.7$ | $8.9 \pm 1.2$ |

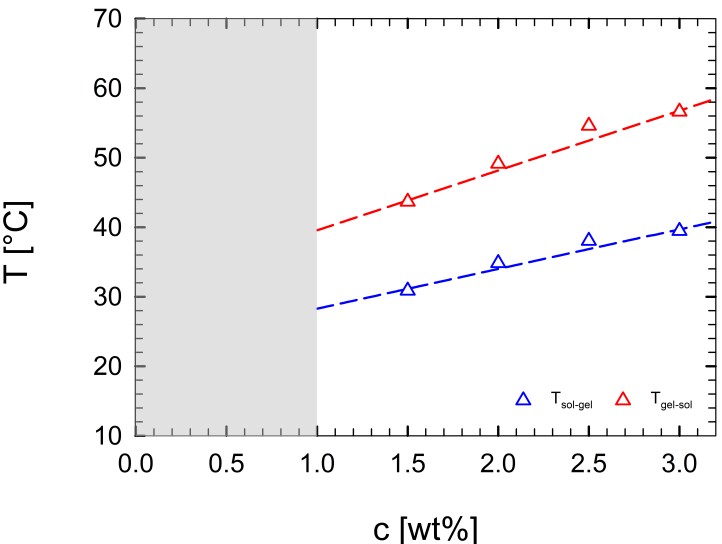

**Figure 5.** $T_{sol\text{-}gel}$ (blue symbols) and $T_{sol\text{-}gel}$ (red symbols) as a function of $\kappa$-c percentage concentration.

Figure 6 displays the gel strength of the $\kappa$-c hydrogels, evaluated as the value of $G'$ at 20 °C, as a function of the biopolymer concentration. Upon increasing the $\kappa$-c concentration, the elastic modulus increases. An increase in the biopolymer content leads, as expected, to the formation of harder gels, due to the decrease in the mesh size of the 3D network [57]. Figure 6 also reports the best fit of experimental data, obtained with a power-law regression via Equation (3):

$$G'(c) = b \cdot c^n \tag{3}$$

where $b$ and $n$ are fitting parameters. The values of these parameters are $b = 121.6 \pm 0.13$ Pa and $n = 4.3 \pm 0.1$. The same power-law trend is also found for other biopolymers such as agarose, gelatin, alginate, and gellan gum [6,58–60]. As an example, the exponent at low agarose concentrations was larger than 4 and was smaller than 2 for higher concentrations [1].

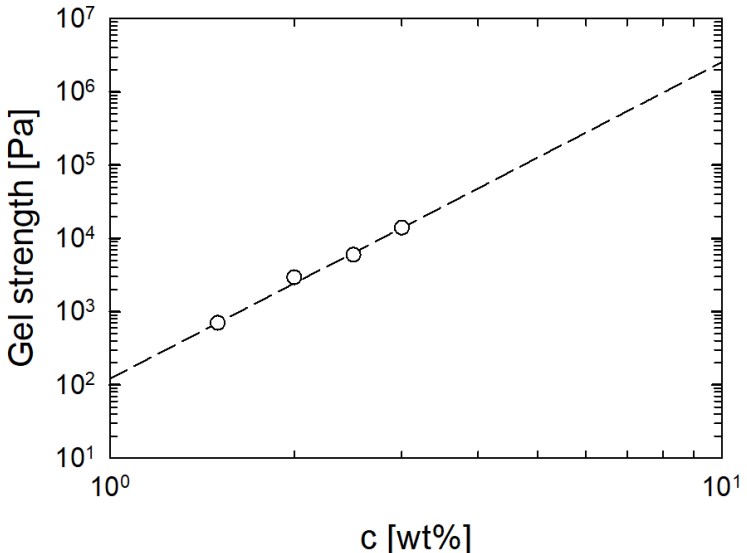

**Figure 6.** Elastic modulus at 20 °C, as a function of $\kappa$-c concentration. Empty circles are experimental data and the dashed line is the power-law regression obtained via Equation (3).

### 3.2. Rheological Behavior of κ-Carrageenan/Xanthan Gum Solutions

In this section, the effect of XG on the gelation properties of κ-c solutions is investigated. For this purpose, the κ-c concentration is fixed at 1.5 wt% and the XG concentration is varied between 0.1 wt% and 1.5 wt%.

Figure 7 reports the temperature dependence of the viscoelastic moduli at 1 °C/min for the lowest (0.1 wt%, Figure 7a), and the highest (1.5 wt%, Figure 7b) XG concentrations. The thermal behavior of the viscoelastic moduli, shown in Figure 7a, is similar to that of water/κ-c solutions, previously discussed. By comparing Figure 7a and Figure 3, it is evident that no difference appears in the ramp response for samples with and without XG. This was confirmed up to an XG concentration of 0.4 wt%.

At higher XG concentrations, the viscoelastic moduli show a significant variation, as demonstrated by Figure 7b, where the temperature response for XG concentration of 1.5 wt% is reported. Figure 7b shows that, at high concentrations of XG, both moduli increase. Moreover, the sample shows an elastic modulus higher than the viscous one, even at high temperatures. This result is due to the high percentage of XG in the solution. According to Wyatt et al. [61], concentrations of XG in water larger than 0.4 wt%, place the solution in the concentrated regime. These solutions exhibit a crossover point in the linear viscoelastic regime at very low frequency ($\approx$0.2 rad/s), indicating that the elastic behavior is dominant. As a result, since the ramps are performed at $\omega = 10$ rad/s, $G'$ is higher than $G''$ in the entire temperature range.

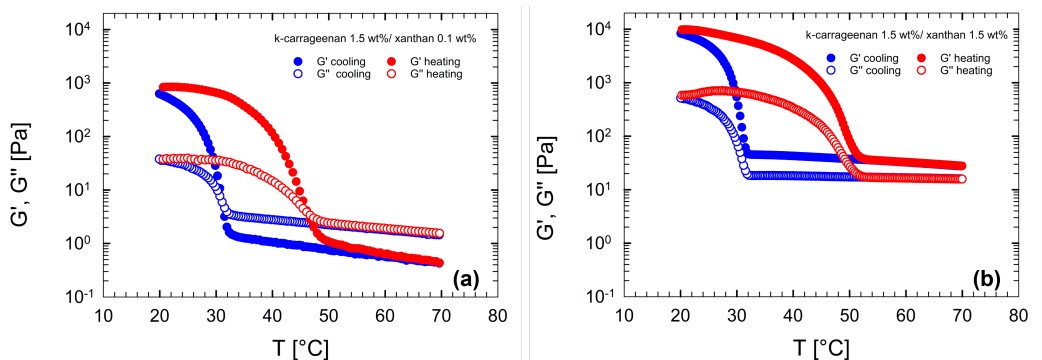

**Figure 7.** Viscoelastic moduli as a function of temperature, for κ-c/XG solutions at 1 °C/min: (**a**) κ-c 1.5 wt% and XG 0.1 wt%, and (**b**) κ-c 1.5 wt% and XG 1.5 wt%.

Figure 8 shows the magnitude of the complex modulus, $|G^*|$, as a function of the temperature, for the investigated κ-c/XG solutions. Figure 8 indicates that the value of $|G^*|$ strongly depends on the XG concentration, both at high and low temperatures. The latter means that the gel strength increases when XG is added. The cooling ramps overlay each other, suggesting that the sol-gel temperature (roughly 30 °C) is slightly affected by the XG concentration. The melting temperatures, on the other hand, increase with XG content.

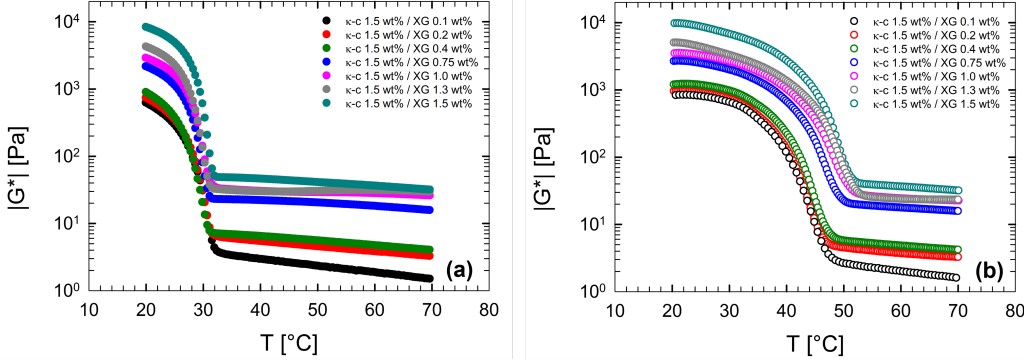

**Figure 8.** Complex modulus as a function of temperature, at 1 °C/min, for solutions with 1.5 wt% of κ-c at different XG amounts. (**a**) Cooling, and (**b**) heating ramps.

A special focus on how the transition temperatures and the elastic moduli vary with XG concentration is provided in Figure 9a,b, respectively. Figure 9a shows that, irrespective of the amount of XG, the gelation temperature can be considered to be constant with XG concentration. The melting temperature weakly depends upon the XG content, and only for concentrations above 0.4 wt%.

Figure 9b depicts both the gel strength of the ternary samples and the value of $G'$ for the XG solutions (without $\kappa$-c), evaluated at 10 rad/s and 20 °C via DFSTs (data not shown), as a function of XG concentration. The gel strength grows significantly with XG concentration, increasing by more than one order of magnitude in a very limited concentration range. The dashed line in Figure 9b is the exponential fit obtained via the following Equation (4):

$$G'(c_{XG}) = \alpha e^{\beta c_{XG}} \tag{4}$$

where $\beta$ is a fitting parameter and $\alpha$ is fixed (714 Pa ± 10 Pa), being the modulus of the pure $\kappa$-c gel. The fitting parameter $\beta$ is 1.7 ± 0.04. The exponential growth in the gel strength upon increasing XG concentration is, as such, apparent. This result can be explained by the fact that the presence of XG enhances the formation of double helices, leading to a stronger gel network. In the presence of water, XG gum exhibits synergistic gelation with $\kappa$-c, as shown in Figure 9b. The synergistic effect is also found for mixed solutions of $\kappa$-c and other polysaccharides [62,63]. We speculate that the increase in both $T_{sol\text{-}gel}$ and gel strength might be related to the excluded volume effect of XG, which leads to the reduction of accessible water molecules, determining an easier $\kappa$-c self-association. This phenomenon results in the formation of "junction zones", with elevated $\kappa$-c concentration. Due to the lack of literature on $\kappa$-c/XG ternary solutions and the absence of additional microstructural data, we are not able to give a definitive mechanism that can explain the formation of a stronger network in the combined presence of XG and $\kappa$-c.

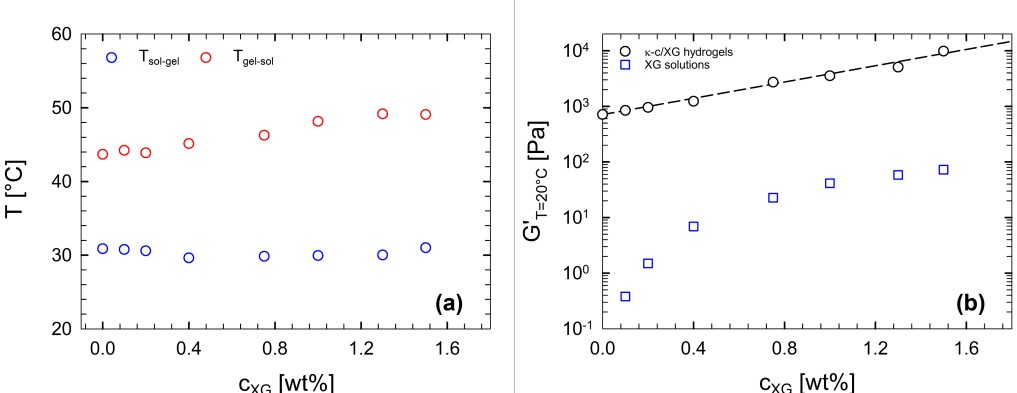

**Figure 9.** (**a**) $T_{sol\text{-}gel}$ (blue circles) and $T_{sol\text{-}gel}$ (red circles), evaluated at 1 °C/min, as a function of the XG percentage concentration. The relative percentage content of $\kappa$-c is fixed at 1.5 wt% for each ternary solution. (**b**) $G'$, evaluated at 20 °C, as a function of the XG percentage in solution. Symbols are experimental data, and the dashed line is a nonlinear regression obtained via Equation (3). For each ternary solution (black circles), the relative percentage concentration of $\kappa$-c is fixed at 1.5 wt%.

### 3.3. Rheological Behavior of κ-Carrageenan/Xanthan Gum Solutions with Sucrose

The results of the DTRTs, performed at 1 °C/min, are shown in Figure 10, where $|G^*|$ is plotted as a function of the temperature and the sugar concentration, for both cooling (Figure 10a) and heating (Figure 10b) ramps. Figure 10 indicates that the complex modulus is strongly affected by the sugar concentration, as the curves are shifted horizontally to higher temperatures, and vertically to higher $|G^*|$ values. The latter marks an improvement in the mechanical properties of the produced gel. Furthermore, the addition of sugar also affects the transition temperatures. In particular, sugar promotes gel formation and

delays its melting. Thus, sugar not only enhances the gel network but also improves its thermal stability.

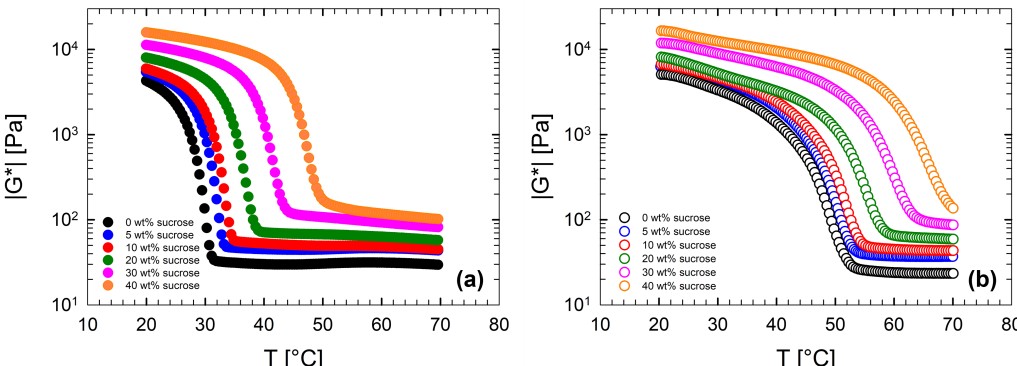

**Figure 10.** Complex modulus as a function of temperature, at 1 °C/min, for quaternary solutions, with a fixed amount of biopolymers (1.5 wt% κ-c and 1.3 wt% XG) and different sugar levels (see legend for details). (**a**) Cooling, and (**b**) heating ramps.

Figure 11a reports the two characteristic temperatures, $T_{sol-gel}$ and $T_{sol-gel}$, as functions of the sucrose concentration. The presence of sugar has a non-negligible effect on both transitions. By comparing the samples at 0 wt% and 40 wt% of sugar concentration, the sol-gel temperature increases by 58%; the gel-sol temperature, on the other hand, increases by 32%. In Figure 11a, the dashed lines are linear fits to the experimental data. $T_0$ is the value of the sol-gel temperature of the solution containing only κ-c and XG. The same approach extends to $T_{sol-gel}$. The values of the fitting parameter, $a$, is $0.42 \pm 0.01$ and $0.37 \pm 0.01$ for $T_{sol-gel}$ and $T_{sol-gel}$, respectively. Both transition temperatures increase linearly with increasing sugar concentration.

Figure 11b reports the gel strength, as a function of the sugar concentration. The strength grows with sugar concentration, following a nonlinear trend. Sugar is expected to play a similar role in the microstructure of both κ-c and κ-c/XG gels.

Yang et al. [64] proposed that, for water/κ-c solutions, the sugar hydration in the bulk phase reduces the average number of available water molecules surrounding the κ-c coils, leading to an enhancement of self-association and formation of junctions [64]. In addition, sugar molecules bind to the helices of the κ-c through the formation of hydrogen bonds, and this facilitates the creation of more and larger double helix aggregates. The incoming gel network becomes stronger, and its mechanical properties are largely improved. We speculate that the behavior proposed by Yang et al. [64] can be applied to our κ-c/XG solutions.

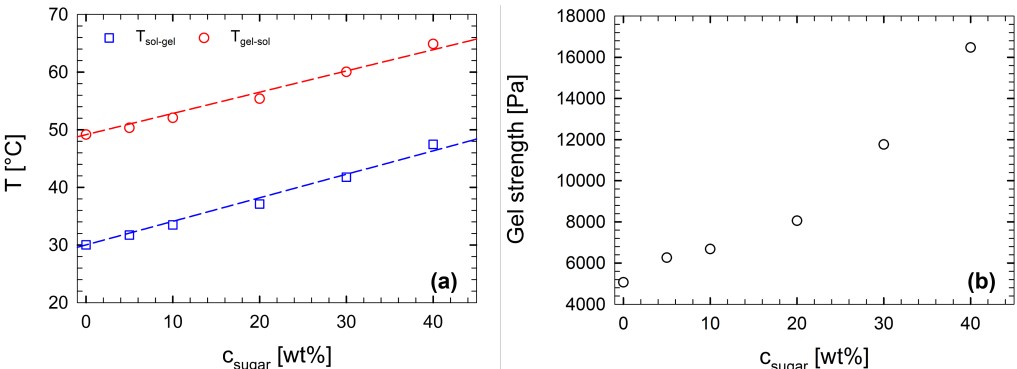

**Figure 11.** (**a**) $T_{sol-gel}$ and $T_{sol-gel}$ and (**b**) gel strength, as functions of the sugar percentage concentration for quaternary solutions, with a fixed amount of biopolymers (1.5 wt% κ-c and 1.3 wt% XG). Dashed lines are obtained via linear regressions.

### 3.4. Yield Stress Evaluation

Figure 12a reports the viscoelastic moduli as a function of frequency, at 20 °C, for three representative samples. They show the classical response of a strong gel, with the elastic modulus practically independent of frequency and much larger than the dissipation modulus [1]. Figure 12b displays the oscillatory stress as a function of the oscillatory shear strain, obtained from a dynamic strain sweep test at 20 °C, for the same samples as Figure 12a. Initially, the stress increases linearly with strain, up to a limiting value, beyond which the samples lose their equilibrium microstructure. The stress and strain critical values can be seen as a measure of the yield behavior of the gel [39,65]. Table 3 reports the yield stress and yield strain values of some samples investigated in this work. The yield strain value appears to be affected solely by the $\kappa$-c concentration, whereas the yield stress depends on both the addition of XG and sucrose.

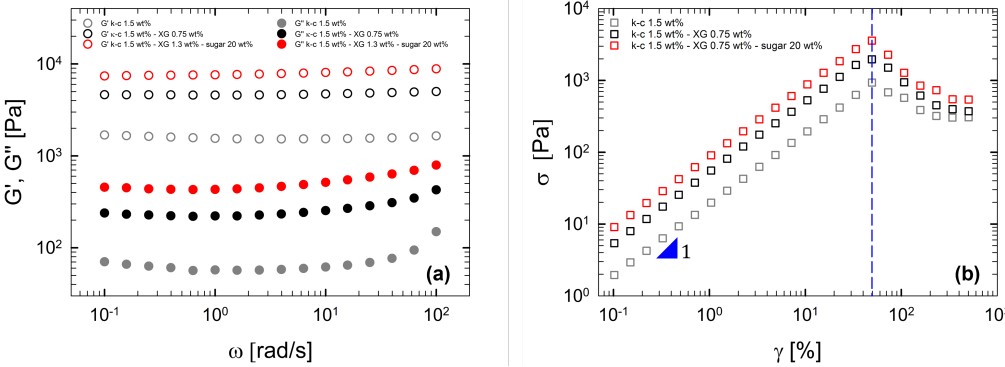

**Figure 12.** (**a**) Viscoelastic moduli as a function of the angular frequency at 20 °C, and (**b**) shear stress as a function of the shear strain at 20 °C. The dashed line indicates the yield point of the gel. Details on samples are reported in the legend.

**Table 3.** Values of yield stress and yield strain evaluated at 20 °C, for selected binary, ternary, and quaternary samples.

| Samples | Yield Stress [Pa] | Yield Strain [%] |
|---|---|---|
| $\kappa$-c 1.5 wt% | 933 | 50% |
| $\kappa$-c 2.0 wt% | 1758 | 34% |
| $\kappa$-c 2.5 wt% | 3042 | 34% |
| $\kappa$-c 3.0 wt% | 4390 | 34% |
| $\kappa$-c 1.5 wt%—XG 0.2 wt% | 1368 | 50% |
| $\kappa$-c 1.5 wt%—XG 0.75 wt% | 1965 | 50% |
| $\kappa$-c 1.5 wt%—XG 1.3 wt% | 2400 | 50% |
| $\kappa$-c 1.5 wt%—XG 1.3 wt%—sucrose 10 wt% | 2450 | 50% |
| $\kappa$-c 1.5 wt%—XG 1.3 wt%—sucrose 20 wt% | 3580 | 50% |
| $\kappa$-c 1.5 wt%—XG 1.3 wt%—sucrose 40 wt% | 5072 | 50% |

### 3.5. Printability of Food Hydrogels

To evaluate the printability of the quaternary solutions with fixed amounts of biopolymers (1.5 wt% $\kappa$-c and 1.3 wt% XG) and different sugar contents, a complex grid was printed (30 mm × 30 mm × 3 mm). As previously shown in Figure 10, such solutions form thermoreversible hydrogels in certain temperature ranges. To carry out a printing process, it is necessary to choose a temperature range where the solutions are in a sol-state, but not too far from the gelling temperature. In this way, these structures can be self-consistent, and the layers merge with each other. According to the data shown in the previous Section 3.3, high temperature extrusion results in the production of structureless filaments, unable to keep the desired shape. Conversely, a low printing temperature means an already gelled system, making it impossible to extrude the solution into a continuous filament. In other words, the printing temperature range is a crucial parameter, because the formulation experiences

a rapid temperature change from a printing temperature in the nozzle cavity to room temperature, when placed on the printer plate. For these reasons, we decided to print our formulations setting the nozzle at $T_{sol-gel}^{onset} + 10\,°C$, which correspond to temperatures of 45, 50, 55, and 60 °C for quaternary solutions containing 10 wt%, 20 wt%, 30 wt%, and 40 wt% of sugar, respectively. Another key parameter is the printing pressure ($P$), which directly affects the resolution of the structures. According to He et al. [66], to maintain fidelity to the 3D model, an ideal printed construct should have a filament width equal to the nozzle diameter. So we have evaluated the resolution of the 3D printed constructs using the nozzle diameter as a reference. Figure 13a displays the filament width obtained by varying $P$ between 4 and 10 kPa, for different food-ink systems. By fixing $P$, the filament diameter increases upon decreasing the sugar content. Irrespective of the sugar concentration, the filament diameter increases upon increasing the printing pressure. The missing data in Figure 13a (4 kPa for 30 wt%, 40 wt%, and 8 kPa for 10 wt%) are due to irregular shapes of the filaments.

Figure 13b reports the optimal pressure value (evaluated at $T_{sol-gel}^{onset} + 10\,°C$), by varying sugar concentration along with the solutions' complex viscosities. As expected, the printing pressure is a parameter strictly related to the formulation viscosity.

Figure 13c–f display snapshots of the frontal area and Figure 13g–j display snapshots of the side area, of structures extruded under optimized pressures (4, 4, 6, and 8 kPa). According to the data shown in Figure 13a, the printed grids obtained with high sugar content (30–40 wt%) show the best shape fidelity. The combination of 1.5 wt% of $\kappa$-c with 1.3 wt% XG and 40 wt% sugar, provides a filament width comparable to the nozzle diameter, which, as suggested by He et al. [66], represents an ideal situation for 3D printed constructs.

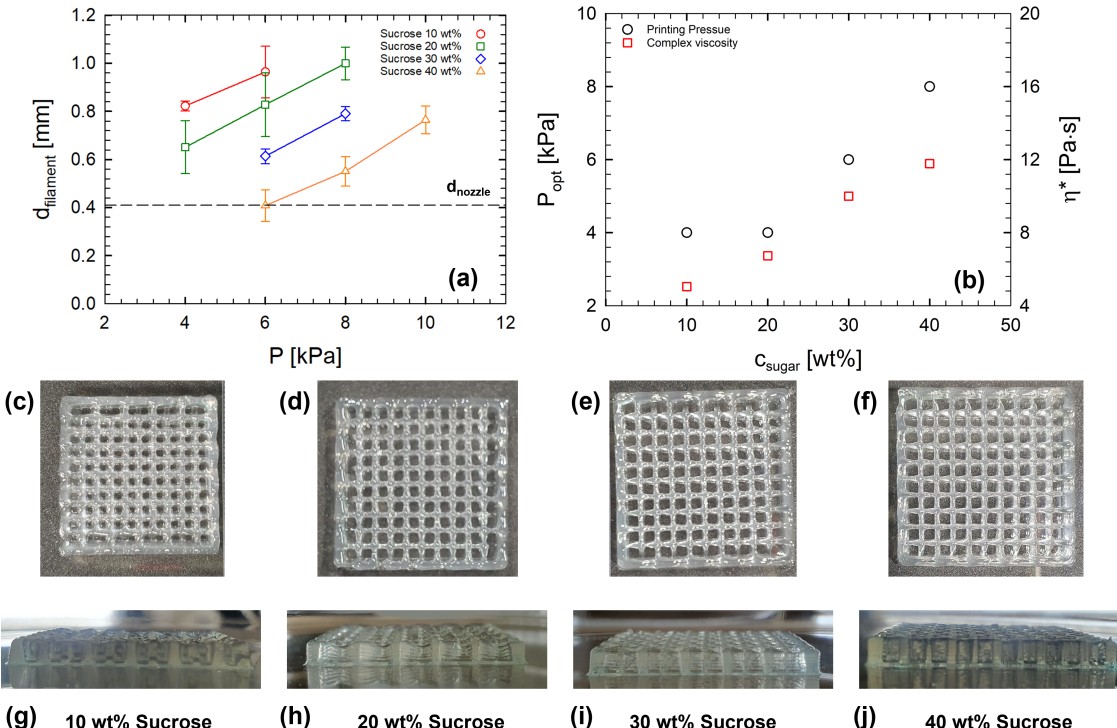

**Figure 13.** (**a**) Filament diameter as a function of the printing pressure, for samples with 1.5 wt% $\kappa$-c and 1.3 wt% XG, at different sucrose levels. Error bars result from different measurements. (**b**) Optimized printing pressure (left y-axis) and complex viscosity, evaluated at $T_{onset} + 10\,°C$ from data shown in Figure 10a (right y-axis), as functions of sucrose concentration, for samples with 1.5 wt% $\kappa$-c and 1.3 wt% XG. Snapshots of the frontal area (**c**–**f**) and side area (**g**–**j**) of the printed grids extruded under best pressure conditions.

## 4. Conclusions

We analyzed the gelation of κ-c solutions, focusing on the semi-dilute regime, with the aim of proposing a suitable green alternative to animal gelatin. A linear dependence of the transition temperatures on the κ-c concentration was found. The gel strength increased, according to a power-law function, with the κ-c concentration. These behaviors were attributed to an increasingly crowded network of helices as the concentration of κ-c grew.

Consequently, compared to gelatin gels, all investigated binary systems had either melting temperatures that were too high, or gel strengths that were too low. For these reasons, we focused on the solution containing 1.5 wt% of κ-c, thus, maintaining the lowest feasible melting temperature. To reinforce the gel network, 1.5 wt% of κ-c solution was mixed with different amounts of XG gum. We demonstrated that, upon increasing XG concentration, the gel strength increased exponentially and the gel-sol temperature slightly increased; the sol-gel temperature, on the other hand, did not change. We suggest that the increase in both $T_{sol-gel}$ and gel strength might be related to the excluded volume effect of XG, which led to the reduction in accessible water molecules, promoting easier self-association of κ-c helices.

As reported, the combination of these two hydrocolloids led to a clear improvement in terms of gel strength. The main weakness of this formulation is an elevated brittleness. We will try to improve this in the future. By fixing the concentration of biopolymers, we evaluated the effect of sucrose on the κ-c/XG solution, finding that both gel strength and transition temperature increased. Due to the ability of the sugar to bind with κ-c via hydrogen bonding, we suggested that the addition of sugar promoted the formation and the aggregation of κ-c helices in a ternary aqueous solution of κ-c/XG and sugar. Further microstructural evidence will shed light on the interaction of XG within the κ-c/sugar network.

We analyzed the printability of 1.5 wt% of κ-c and 1.3 wt% XG, by varying the sugar concentration from 10 wt% to 40 wt%. We proved that high sugar concentrations improve the printability and the resulting shape fidelity. Nevertheless, a quantitative evaluation of additional printing parameters, such as the adhesion properties, is necessary in the future, to find direct applications in various industrial fields.

**Author Contributions:** P.R.A. and S.R.S. contributed equally to this work. Conceptualization and methodology, P.R.A., S.R.S., R.P. and N.G.; formal analysis, R.P.; investigation, P.R.A., M.G.E. and S.R.S.; resources A.S. and M.D.; data curation R.P. and N.G.; writing—original draft preparation P.R.A., S.R.S. and R.P.; writing—review and editing P.R.A., S.R.S., R.P. and N.G.; supervision R.P., S.A. and N.G. All authors have read and agreed to the published version of the manuscript.

**Funding:** This research received no external funding.

**Institutional Review Board Statement:** Not applicable.

**Informed Consent Statement:** Not applicable.

**Data Availability Statement:** The data that support the findings of this study are available from the corresponding author upon request.

**Acknowledgments:** R.P. acknowledges TA Instruments, for awarding the rheometer used in this study as part of the "Distinguished Young Rheologist" program. P.R.A. acknowledges the support of the Italian Ministry of University, project PRIN 2017, 20179SWLKA "Multiple Advanced Materials Manufactured by Additive technologies (MAMMA)".

**Conflicts of Interest:** A.S. and M.D. are employees of Perfetti Van Melle. They both state that in this paper there is nothing that may be considered to be a conflict of interest.

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
