# Peer review of "Thermorheological Behavior of κ-Carrageenan Hydrogels Modified with Xanthan Gum"

_fluids, doi:10.3390/fluids8040119_

Round 1

Reviewer 1 Report

In this manuscript, Avallone et al. propose k-carrageenan as an alternative for reproducing the gelatin features of animal gelatin gels. Through carefully executed bulk rheological experiments, the authors present data on the thermorheological response of  three materials, namely (i) k-carrageenan solutions at various concentrations, (ii) mixture of k-carrageenan and xanthan gum at various concentrations, and (iii) mixture of xanthan gum, k-carrageenan and sucrose solutions at various sucrose wt%. The authors present data on viscosity vs shear rate, effect of temperature on G’ and G”, effect of heating/cooling ramps on sol-gel and gel-sol transition, and effect of material concentration on the gel strength. The authors compare results from these cases and provide reasoning on the observed trends.

Overall, this work is very impressive. The experiments are well designed, and the paper is written clearly and easy to follow.   In the current version, manuscript is, however, not suitable for Fluids journal. The paper can be considered for publication in Fluids after the follows concerns are addressed:

 COMMENTS

1.     I think an important control missing in the current manuscript is the data on yield stress as a function of k-carrageenan concentration, data on yield stress as a function of xanthan gum concentration, and also on the wt% of sucrose. I think if that data is included in paper, it would complete the rheological response of all three materials considered in this manuscript.

2.     How is the sol-gel and gel-sol transition temperature determined? Is it just a visual determination?

3.     The authors present interesting rheological data in this manuscript. However, the paper falls short when it comes to discussing k-carrageenan as an alternative to animal gelatin. Based on the results from this paper, can the authors add a paragraph in conclusion that discusses what are the gains/loss in properties if one were to replace animal gelatin with k-carrageenan.

Reviewer 2 Report

The article by P. R. Avallone et al. investigates the thermorheological behavior of k-carrageenan hydrogels modified with xanthan gum. The study shows that there is a linear dependence of the transition temperatures on the k-c concentration, and that the gel strength increases according to a power law function with the k-c concentration. The article is clear, concise and logical, deals with a relevant area of research, and can be accepted for publication in Fluids.

The article has the following comments:

1. For Figure 1.(a), Illustrate that the viscosity of dilute and semi-dilute solutions is measured over a range of different shear rates.

2. For Figure 3, explain that the inflection points on the G' and G'' curves are related to the gelation time of the sample.

Reviewer 3 Report

Article: Thermorheological behavior of k-carrageenan hydrogels modified with xanthan gum.

The main purpose of this work is to display a green hydrocolloid network able to reproduce the gelation features of animal gelatin gels.

This topic is considered relevant since hydrocolloids is very used mainly in food industries.

There are some specific comments listed below:

Page 3:

The methodology is well described and includes all experiments carried out during the research. However, it is necessary, when applicable, to include the references of items 2.2 to 2.5.

It is known that statistical analysis provides a correct interpretation of experimental data. So, statistical analysis was not mentioned in the experiments of this work, please clarify.

Results:

Figures 3 to 6 need to be formatted, according to other figures in the article.

Page 7:

Table 1 identification needs to be moved to before this table.

Page 8:

Table 2 identification also needs to be moved to before this table. However, it is necessary to change this nomenclature. The correct identification of Table 2 is:

Table 2. Values of the regression parameters of the linear fits in Figure 5.

Identification of Figure 5 is shown below:

Figure 5. Tsol-gel (blue symbols) and Tgel-sol (red symbols) as function of k-c percentage concentration.

The following part needs to include in the text of article: “The transition temperatures are evaluated at a cooling/heating ramp rate of 1C/min. Dashed lines are linear regressions. The gray region identifies a concentration range in which no gelation occurs”.

Page 13:

Discussion of the results in Figure 12 is confusing. Also, the letters “g” through “j” in this figure are not visible. Please rewrite this figure and its discussion of results.

It has been discussed that the impression pressure (P) directly affects the resolution of these structures. It is not clear which pressure range was used in this work. Also include this information in the methodology (item 2.5).

Conclusion

The authors correctly indicate, how the results are related to the studies.

Round 2

Reviewer 1 Report

The authors have addressed my concerns. I recommend publication

Reviewer 3 Report

The main objective of this work is to present a green hydrocolloid network able to reproduce the gelling features of animal gelatin gels, and as I mentioned before, this topic is considered relevant for the area in which it insert.

The authors answered to all specific comments listed before.